# Family Support, Resilience, and Life Goals of Young People in Residential Care

**DOI:** 10.3390/bs14070581

**Published:** 2024-07-09

**Authors:** Cristina Peixoto Alves, Inês Carvalho Relva, Mónica Costa, Catarina Pinheiro Mota

**Affiliations:** 1Department of Education and Psychology, University of Trás-os-Montes and Alto Douro, Quinta de Prados, 5000-801 Vila Real, Portugal; alves.peixoto@hotmail.com (C.P.A.); irelva@utad.pt (I.C.R.); monicac@utad.pt (M.C.); 2Center for Psychology at the University of Porto (CPUP), Rua Alfredo Allen, 4200-135 Porto, Portugal; 3Research Center in Sports Sciences, Health Sciences and Human Development (CIDESD), 6201-001 Covilhã, Portugal; 4Centre for Research and Intervention in Education (CIIE), 4200-135 Porto, Portugal

**Keywords:** family support, life goals, resilience, life projects, young, residential care

## Abstract

Establishing goals for young people in residential care (RC) is a gap in the literature, especially in terms of the relationship between family support and resilience. The literature suggests that RC is associated with the breakdown of family relationships, so the possibility of the family playing a positive role in establishing life goals for young people is reduced. However, family support in the context of organization and stability can be assumed to be a protective factor for the formulation of life goals and contribute to the resilient development of young people. This study aimed to analyze the role of family support in the process of setting goals for young people in RC, as well as to analyze the potential mediating role of resilience in the previous association. The sample included 124 young people aged between 12 and 23 years living in RC. The results point to a positive association between family support (autonomy) and the planning of life goals and verify the total mediating effect of resilience in this association. The results are discussed considering attachment theory and the role of the family in the adaptive development of young people. This study’s findings provide important indications for developing future interventions.

## 1. Introduction

According to attachment theory, the quality of the affective relationship between children and primary caregivers is an important indicator of development over time [1]. Establishing stable, safe, and close bonds with parental figures promotes the development of trust and security in relationships with children and young people [2]. In this way, depending on the responsiveness of caregivers, children internalize a set of expectations about themselves, others, and the surrounding world, building dynamic internal models that serve as cognitive and affective maps to interpret past experiences and guide future actions [2,3]. Thus, the elaboration of positive dynamic internal models in individuals triggers more optimistic views of their skills, enabling them to adapt to adversity [1,4,5]. However, because of their history of family separations and insecure/disorganized attachments, young people in residential care (RC) often face challenges in these processes [6]. Although frequently marginalized by RC professionals, family support and intervention can represent an important opportunity for affective reorganization and contribute to the (re)construction of family bonds capable of enhancing the adaptation and positive development of young people [7,8].

Most children and adolescents living in RC contexts have been exposed to a variety of adverse family experiences [9]. However, despite being involved in neglectful and mistreating family dynamics, many young people in RC tend to value and prioritize family ties, seeking contact and closeness with primary caregivers [10,11]. Although there is some controversy, the literature indicates that the family’s involvement in RC reflects greater placement satisfaction, facilitates adaptations, fills prolonged stays, and contributes to the realization of life projects (LPs) [12,13]. LPs refer to individual and personalized plans determined by the child protection system, focusing on young people’s personal, social, and educational development in RC [14]. These plans assume various typologies; however, “reintegration into the nuclear family” and “autonomy” are among the most prevalent in the Portuguese system of protection [14]. The involvement of caregivers in the definition of LPs is related to developing feelings of security, belonging, and adaptation, which can lead to the design of life goals [15,16,17,18]. However, in the context of RC, there must be significant investment in working with families [7,19] to the extent that a large part of family dynamics is marked by some disorganization and instability that does not provide emotional security to young people. In the presence of inconsistent family relationships, guided by negligent and discontinuous care, which tend to result in less optimistic perceptions of family support [20], distance from the family can contribute to young people’s regulation and internal organization [21].

According to the literature, family support refers to the manifestations of support promoted by trust, dialogue, affection, care, protection, and mutual help among different members of a family structure [22]. Sex differences seem to be pointed out by the research regarding family support. Some studies indicate that females perceive more significant support from primary caregivers [23]. However, studies carried out in the context of RC suggest higher perceptions of affectivity, closeness, and family inclusion associated with males [24]. The literature has widely supported the consideration of parental nuclei as a system of protection against adversity [4,25]. A systematic review by Meng et al. [26] highlighted that resilience development can encompass various personal attributes and systemic resources, of which positive parenting and parental availability and care are particularly relevant. From this perspective, family relationships can mitigate the impacts of adversity on youth development and well-being and contribute to the construction of personal resources that enhance post-traumatic growth [25]. Security and availability in parental relationships can also contribute to acquiring feelings of security and stability, resulting in the positive adaptation of young people in RC [27,28,29]. However, given the circumstance of family instability and lack of contact with primary caregivers, other alternative figures may contribute to the emotional stability and resilient adaptation of young people in RC, namely, teachers, school staff, and careworkers of the institution [5,28,29].

In addition to fostering resilience, careworkers can influence young people’s expectations, ideas, choices, and goals [15,17,30]. The development of life goals reflects an interaction among personal, social, and relational factors [31,32], resulting in a motivational process through which individuals direct effort and dedication, as well as execute plans to achieve desired results [33]. Although research on life goals is scarce in the context of RC, it is known that the cumulative effect of adverse experiences seems to deprive young people of the resources and opportunities they need to select, explore, and plan life trajectories [8,34]. In this sense, goal-building can be challenging for young people in RC [17]. Secure parental relationships, guided by emotional support, can provide feelings of self-worth and competence, resulting in a more promising design of life goals [1,15,17,35]. In addition, the autonomy provided by parents also seems to be associated with the commitment of young people to the design and achievement of goals [36,37,38,39]. However, experiences of abusive, threatening, or negligent relationships with biological caregivers can enhance defensive orientations and compromise security in the construction of life trajectories [15,40].

According to Bronfenbrenner’s bioecological model, human behavior and development are shaped by the interaction among different ecological systems [41]. In this context, along with family support, the goal-setting process seems to be influenced by individual and contextual variables, such as sex and length in RC [15,42,43]. Regarding sex, even if the research is ambivalent [42,44], previous studies indicate that young males tend to be more actively involved in goal-setting processes [15,45], essentially image-driven, financially successful, popularity, and lifestyle [43,44]. Despite the controversy in the results reported by the scientific community, gender effects on the behavioral adjustment of adolescents living in RC have been found [46]. Boys tend to adopt more reactive, violent, and deviant behaviors compared with girls; on the other hand, girls tend to report more internalizing problems such as depression and anxiety, which, in both cases, can affect the youth’s adaptation and performance setting [23,47,48,49]. The scientific community has sparsely explored the role that length in RC plays in the construction of LP, and it needs to be more consistent. A study developed by Mota et al. [43] found negative associations among the length of stay, age of entry into the current household, and extrinsic life aspirations of 296 Portuguese adolescents living in RC. Also, in a study conducted with young people in RC, Mota et al. [50] found that young people who spent more than ten years in RC have the most anxious attachments. On the other hand, Costa et al. [46], in their study carried out in Portugal with young people in RC, pointed out that the length of foster care is significantly correlated with less emotional stress, so other variables, such as cohesion in RC, may be relevant for the adjustment of young people. Other studies do not confirm the significance of the length of RC in the development of personal variables such as self-efficacy [51] or relational dynamics with caregivers in the RC home [52]; this reinforces the hypothesis that the length of RC itself is not decisive in the experience of young people, but rather its combination with personal and contextual factors.

Recent studies, although scarce, found associations between resilience and the elaboration of life goals [8]. Resilient individuals tend to have higher self-esteem and perceptions of greater self-efficacy and self-concept [1,53,54]. Resilience in young people also seems to be related to coping, motivation, and self-regulation strategies [54,55]. Thus, as a result of their internal resources, resilient young people are able to translate greater security for the projection of personal achievements in the future [56], which may lead to their greater involvement in the design and accomplishment of life goals [53,57]. Therefore, considering that young people in RC experience inconsistent care and affective discontinuities in the family [6,43], but, at the same time, continue to feel a sense of belonging to primary caregivers, it is important to work on family dynamics with young people and the institution in order to promote parental involvement and re-education for young people’s adaptive life projects [7,8].

The present study aims to fill gaps in the literature, especially regarding family support and establishing future goals for young people in RC. In addition, it seeks to understand the contributions of the resilient process in this association and highlight the role of the time of reception, the age of entry into RC, and the typology of LPs in establishing life goals. It is important to note that, given the scarcity of studies on the subject, some of the hypotheses formulated during this study are exploratory.

## 2. Materials and Methods

### 2.1. Participants

The sample included 124 young people living in RC facilities, comprising 75 males (59.5%) and 49 females (39.5%) aged between 12 and 23 years (*M* = 16.71; *SD* = 2.43). The age these young people entered in their current RC ranged from 2 to 20 years (*M* = 13.10; *SD* = 3.63), and the length of stay ranged from less than 1 month to 192 months (*M* = 40.10; *SD* = 41.41). It should be noted that a quarter of the sample had previously attended other RC facilities. Regarding LP, almost all participants (96.8%) contemplated a defined LP, with 79 (67.7%) autonomy plans and 43 (34.7%) (re)integration projects into the nuclear family. Only a minority proportion were part of civil sponsorship and (re)integration projects in the extended family (0.8%). About 2 out of 3 young people (64%) assumed their families’ participation in LPs. Overall, 23.8% missing was reported for the family’s involvement in LPs. It should be noted that the institutions participating in the present study did not include specialized/therapeutic RC. These adolescents lived in a residential care institution because of a diverse set of adverse life situations, namely, parental neglect or lack of family socio-economic conditions. The participating residential care institutions did not include children with mental disabilities/disorders or deviant behaviors (conduct disorders or substance abuse). The sample included in this study was homogeneous in relation to race and ethnicity.

Although other significant adult figures may be relevant to young people who experience affective discontinuities (e.g., friends, godparents, coaches), in this study, we emphasize the role of the family, which usually maintains contact with young people and provides more attention to the emotional involvement of young people.

### 2.2. Measures

Sociodemographic questionnaire. A sociodemographic questionnaire was designed to collect sociodemographic information relevant to the characterization of participants (e.g., sex, age). In addition, it allowed access to data alluding to the institutionalization process of young people (e.g., age at admission to RC, length of stay in RC, type of LP, and family involvement in the LP). Although the questionnaire was explicitly aimed at young people, the collaboration of RC’s case managers/technical directors was occasionally requested.

Family Support Perception Inventory (FSPI). (Baptista [58].) This self-report questionnaire consists of 41 items that assess the perception of family support in the young population. It considers three dimensions including affective-consistent (22 items), adaptation (11 items) and autonomy (8 items). The Likert response scale for each item ranges from (0) “*almost never or never*” to (3) “*almost always or always*”. To adapt the instrument to the previously determined objectives, we decided to use only the adaptation dimensions, which evaluate the absence of negative feelings and behaviors towards the family, such as anger, isolation, exclusion, shame, misunderstanding, and lack of interest (e.g., “*the members of my family only think of themselves*”), and autonomy, which measures family relationships of trust, freedom, and privacy (e.g., “*my family accepts me as I am*”). It should be noted that because they were described negatively, all items of the adaptation subscale were scored inversely. Psychometric studies have revealed adequate Cronbach’s alpha coefficients for each dimension including adaptation (α = 0.905) and autonomy (α = 0.940). The confirmatory factor analysis (CFA) presented acceptable adjustment indices, χi^2^ (147) = 295.441; *p* < 0.001, χi^2^/gL = 2.010; CFI = 0.911; TLI = 0.897; RMR = 0.0628, and RMSEA = 0.091.

Short Self-Regulation Questionnaire (SSRQ). (Carey et al., [59], Portuguese adaptation by García Del Castillo and Dias, [60].) This questionnaire, characterized as a self-report instrument, aims to assess the ability of individuals to self-regulate their behaviors and plan their self-determined goals. It covers 16 items positively described and divided into three dimensions, including goal setting, decision-making, and learning from mistakes. It should be noted that to meet the underlying objectives of this study, we chose to use only the objective setting dimension (e.g., “*I usually track my progress until I reach my goals*”), according to the Personal Agency model proposed by Nunes et al. [61]. The answer options for each item are distributed according to a 5-point Likert scale, ranging from (1) “*strongly disagree*” to (5) “*strongly agree*”. For the present sample, the dimension revealed a Cronbach’s alpha of α = 0.844 and good adjustment indicators, χi^2^ (34) = 73.117; *p* < 0.001, χi^2^/Gl = 2.152; CFI = 0.907; TLI = 0.877; RMR = 0.0648 and RMSEA = 0.097.

Resilience Scale (RS). (Wagnild and Young [62], Portuguese adaptation of Gonçalves and Camarneiro, [63]). This self-report questionnaire examines the adaptive capacity of young people in the face of adverse events. It consists of 25 items, distributed in two dimensions, namely, acceptance of self and life (ACCEPSL) (14 items), which assesses how accepting young people are of themselves and their experience (e.g., “*I feel proud to have achieved goals in my life*”), and personal competence (PCOMP), which refers to their perception of competences (11 items) (e.g., “*When I make plans, I see them through to the end*”). All items are formulated in a positive sense and arranged on a 7-point Likert scale, ranging between (1) “*totally disagree*” and (7) “*totally agree*”. In the scope of this study, it was decided to group resilience into a single construct, aiming to achieve more appropriate levels of internal consistency and adjustment indicators [64]. However, it should be noted that the one-dimensional structure of the instrument required the removal of 4 items (11, 20, 22, 25). The internal consistency analysis for the total scale, using Cronbach’s alpha, was 0.911. For the present sample, the CFA of the one-dimensional instrument revealed an adequate fit of the model, χi^2^ (187) = 312.558; *p* < 0.001, χi^2^/gL = 1.672; CFI = 0.866; TLI = 0.850; and RMR = 0.069 and RMSEA = 0.074.

### 2.3. Procedures

The University of Trás-os-Montes and Alto Douro Ethics Committee approved the present study. The project was presented to the technical team of each RC, who were asked for permission to collect data, and the objectives and practical implications of the study were clarified. Before administering the questionnaires, a spoken reflection was carried out, which sought to assess the protocol’s formal and semantic comprehensibility and adequacy. The recruitment of participants was carried out considering their interest and availability to participate and did not involve monetary compensation. All participants signed an accessible and informed consent form. The research procedures followed the Code of Ethics and Deontology of the Order of Portuguese Psychologists, safeguarding voluntariness, anonymity, and confidentiality assumptions. Data collection took place after work and in a face-to-face and collective format. The principal investigator supervised the protocol administration to clarify this study’s objectives and answer possible questions. The research protocol self-report questionnaires were randomly inverted to avoid bias in the answers provided because of the fatigue factor.

### 2.4. Data Analysis

The present research assumed a quantitative and cross-sectional methodology. Data were processed using statistical programs *SPSS—Statistical Package for Social Sciences* (version 26.0) and *AMOS—Analysis of Moment Structures* (version 29.0) for Windows. Preliminary analyses included the identification and exclusion of missing data and potential outliers. The normality of the distribution of the collected data was tested and confirmed, and parametric tests were used. The factorial structure of the instruments was evaluated using 1st-order CFA. The following procedures included descriptive (means and standard deviations), correlational (*Pearson*’s correlations), and univariate variance (*t*-tests) analyses. Finally, given the objectives outlined, mediation analyses were carried out to determine the indirect effects of the resilience variable on the association between family support and the elaboration of life goals. The typology of LPs (0 = life autonomy plan; 1 = other), the age of entry into the current RC, and the length of stay in the institution were included as covariates in the model, controlling for their role in the study-dependent variable. Correlational analyses assumed reference values between 0.10 and 0.29 for weak associations, 0.30 and 0.49 for moderate associations, and above 0.50 for strong associations [65]. The cohort points for acceptable fit values were CFI and TLI ≥ 0.90 and RMSEA and RMR < 0.10 [64]. All results were analyzed and interpreted from a significance level of <0.05.

## 3. Results

### 3.1. Correlation among Family Support, Resilience, and Goal Setting

The results revealed significant associations between the dimensions of family support and the goal-setting process. There was a significant, positive, and moderate correlation with the autonomy dimension (*r* = 0.307, *p* < 0.01), as well as a significant, positive, and weak correlation with the adaptation dimension (*r* = 0.274, *p* < 0.01). Regarding the association between the dimensions of family support and resilience, there were significant, positive, and low correlations with the autonomy dimensions (*r* = 0.269, *p* < 0.01) and adaptation (*r* = 0.253, *p* < 0.01). Finally, given the association between resilience and the goal-setting process, a positive, significant, and high correlation was observed (*r* = 0.631, *p* < 0.01) (Table 1).

### 3.2. Variance in Family Support, Resilience, and the Goal-Setting Process According to Sex and Family Participation in LP

The analysis of variance of family support according to sex pointed to statistically significant differences in the adaptation dimension *t* (122) = 2.447, *p* = 0.016, IC95% [0.006; 0.629]. It was found that young males show higher levels of adaptation compared with young females (*M* = 2.97; *SD* = 0.739). On the other hand, for the autonomy dimension, *t* (122) = 0.956, *p* = 0.334, IC95% [−0.156; 0.447], there were no statistically significant differences in sex. Regarding resilience, statistically significant differences were observed according to sex *t* (122) = 2.447, *p* = 0.016, IC95% [0.006; 0.629], where males (*M* = 5.54; *SD* = 0.897) presented higher levels of overcoming adversity when compared with females (*M* = 5.06; *SD* = 1.056). Finally, regarding the variance in the goal-setting process according to sex, statistically significant differences were observed, *t* (122) = 2.776, *p* = 0.006, IC95% [0.111; 0.664]. The results suggested that young males show greater involvement in the design of future goals (*M* = 4.15; *SD* = 0.785) compared with young females (*M* = 3.77; *SD* = 0.718) (Table 2).

Regarding the differential analysis between family support and family participation in LPs, there were statistically significant differences in the adaptation dimension *t* (54.11) = 4.23, *p* < 0.001, IC95% [0.380; 1.067]. The results showed that young people whose families participate in their LPs provide more excellent family support in terms of adaptation (*M* = 3.46; *SD* = 0.631) compared with young people whose families are not involved in the definition of their LPs (*M* = 2.73; *SD* = 0.891). However, no statistically significant differences were found in the autonomy dimension, *t* (93) = 1.372, *p* = 0.173, IC95% [−0.108; 0.589], given the family’s participation in the LP. Regarding resilience, univariate analyses showed statistically significant differences, *t* (93) = 2.446, *p* = 0.016, IC95% [0.091; 0.875]; according to the family’s participation in the LP, young people whose families are involved in LPs reported higher resilience rates (*M* = 5.65; *SD* = 0.933) than young people whose families are not involved in LPs (*M* = 5.17; *SD* = 0.921). Finally, regarding the variance in establishing objectives according to the family’s participation in the LP, statistically significant differences were observed, *t* (93) = 2.196, *p* = 0.031, IC95% [0.035; 0.700]. Thus, young people whose families are involved in the design of their LPs have more significant established goals (*M* = 4.17; *SD* = 0.764) when compared with young people with non-participating families (*M*= 3.81; *SD* = 0.824) (Table 3).

### 3.3. Mediating Effect of Resilience on the Relationship between Family Support (Autonomy) and Goal Setting

A structural equation model was developed through *AMOS* using the bootstrap technique [64]. A *Path Analysis* was conducted to test the mediation among the predictor (family support), mediator (resilience), and dependent (goal setting) variables. The results of the analyses verified that the autonomy dimension of family support positively predicted the process of goal setting (β = 0.23). On the other hand, the adaptation dimension did not reveal predictive effects for any of the variables under study; thus, it was removed [64]. In the final model, the association between adaptation and goal setting lost significance after introducing the mediating resilience variable. Thus, a positive total mediation was observed, where resilience mediated the association between autonomy and goal setting (*p* < 0.001, β = 0.68, IC 90% [0.113, 0.346]). The variables including age at entry into RC, length of stay, and type of LP were controlled in the model to ascertain their role in setting objectives. It was found that the length of stay in RC (*p* = 0.015, β = −0.23) was negatively correlated with the elaboration of life goals of young people in RC. On the other hand, there were no significant contributions by age of entry into RC or the type of LP to setting objectives. The final model showed good adjustment ratios as follows: [χi^2^ (48) = 73.616; *p* = 0.010, χi^2^/gL = 1.534; CFI = 0.964; TLI = 0.951; RMR = 0.066 and RMSEA = 0.068] (Figure 1).

## 4. Discussion

The main objective of the present study was to analyze the associations between family support and goal setting by young people in RC and investigate the mediating role of resilience in the association. In addition, this study aimed to clarify the role of the time of reception, the age of entry into RC, and the typology of LP in building the life goals of young people in RC.

The results revealed positive and significant associations among family support, resilience, and the process of elaborating life goals. This result is in line with what is expected and is supported by the Bioecological Model of Bronfenbrenner [41]. This model emphasizes the interdependence among contextual factors, where it is possible to frame family support and intrinsic characteristics of the individual, such as resilience and the processes of planning and building life goals [41]. According to the literature, the interpretation of the family as a welcoming, safe nucleus capable of favoring the construction of intense and persistent relationships results in the greater involvement of young people in constructing concrete and meaningful goals [17,30,37]. However, it is known that a high proportion of the population in RC experiences family breakdowns, as well as parental relationships based on inconsistent care and insecure attachment patterns, which seem to compromise the projection of young people in the future [6,32]. Therefore, the conclusions of the present study highlight the importance of designing and implementing programs aimed at families capable of meeting young people’s affective and relational needs and enhancing their adaptive development [7,8]. In conclusion, the results stress the importance of having a “strengthened family will be able to fulfil its functions and tasks, which will, in turn, contribute to the strengthening of individual family members as well as the community in which family is living” [4] (p. 15).

Regarding the variance analysis, there were statistically significant differences in family support regarding sex. The results highlighted that young males perceived higher family support (adaptation) than females, which may be related to higher perceptions of support, belonging, and family inclusion. Despite the controversy in the results reported by the scientific community, studies support the idea that boys and girls tend to perceive support from their families differently [23,24]. This evidence seems to be in line with traditionally assigned gender roles and also with the differences in proximity-seeking by individuals of both sexes, whereby proximity-seeking is more prominent in females [47]. According to Parra et al. [66], females value seeking emotional and relational connections within family dynamics and establishing close, affective, and intimate interactions with primary caregivers. However, in the context of RC, girls are susceptible to relational breakdowns [67], interpreting lower feelings of support when distancing from their families of origin [48]. In contrast, the support perceived by males tends to take on a more instrumental nature, focused on prioritizing personal care and maintaining material resources [68,69]. Thus, the results of the present study may underlie the prioritization that boys in RC tend to attribute to pragmatic aspects of family support.

In addition, there were statistically significant differences in resilience according to sex and, once again, it was found that males showed greater resilience. These results align with previous studies that show differences in skills and internal resources for overcoming adversity between sexes [49,63]. Thus, in the present sample, young males seem to demonstrate a more effective ability to deal with controversial situations, exhibiting a greater propensity to accept their experiences and confidence in rebuilding their own lives [70,71]. However, given the inconsistency in the results disseminated by the scientific community [72], some caution is necessary in the interpretation of this result, mainly because the resilience variable in the sample of young people in RC may contain some exacerbation because of possible defensiveness or social desirability.

Also, there were significant differences in establishing goals according to sex, where young males tended to be more actively involved in designing life goals than females. Some studies show that male adolescents place greater importance on image-oriented life aspirations, financial success, popularity, and lifestyle [43,44]. Conversely, females tend to have higher academic/professional and family expectations [32,42]. According to Huang et al. [45], the high demands and fewer opportunities females face, particularly at work, may hinder their life aspirations. In this sense, and given the lack of unanimity in the literature on life goals, the results of the present study seem to be justified by cultural and gender stereotypes, which tend to be attributed to males’ greater proactivity and motivation in structuring goals [73,74]. In addition, the differences in opportunities, social valuation, and participatory possibilities between men and women can contribute to the more active involvement of young males in the projection of personal achievements in the future. Mota and Oliveira [15] corroborated this observation in a study conducted with adolescents in RC, where it was found that males had higher life goals, particularly in terms of satisfaction with life.

The analysis of differences in family support compared with family participation in LP points to differences in the adaptation dimension, so young people whose families are involved in LPs have higher family support (at the level of adaptation) compared with young people whose families are not involved. To this extent, the results indicate that the participation of families in the intervention processes of RC facilities provides young people with greater feelings of inclusion, closeness, and support, which can culminate in their acquisition of more adaptive strategies. The literature needs to be more extensive on family involvement in RC [7]. However, studies have shown that the participation of parental figures in the child and adolescent protection system, especially in the design of LP, can promote the achievement of family reunification [7,12]. Thus, contact between young people and their parents or primary caregivers constitutes the basis for the maintenance and development of family relationships so that the work of each case’s personal and contextual dimensions can re-establish emotional response skills and parenting strategies in caregivers [75]. In support of the results of the present study, Balsells Bailón et al. [12] concluded that the participation of parents in foster care seems to exacerbate their perceptions of self-sufficiency, competence in care, and satisfaction in parenting, making these caregivers more aware of their failures, which may increase their availability to support their children. Therefore, it is suggested that whenever there are conditions in the reception measures of each young person, assiduous family participation in RC, guided by availability and concern with the intervention process, can promote the re-establishment of parental ties and, in turn, achieve LPs in a faster and more effective way.

In addition, there were differences in resilience regarding family participation in the LP. These results are consistent with previous studies and indicate that for young people in RC, their families can act as important sources of support and emotional support capable of driving resilient and adapted trajectories [28,29,76]. Thus, young people whose families participate in their LPs accept and consider their current and past experiences as potential opportunities for learning and personal growth, gaining greater confidence in mobilizing resources to face adverse situations. Although limited and controversial, the literature has recommended the need for greater involvement of families in the intervention processes during RC placements to the extent that parental nuclei can boost the resilient development of young people through support, acceptance, and consistency in care [27,28,29]. Considering Pinheiro et al.’s [28] position that past trauma experiences cannot be “changed” (p. 820), it is important to develop parents’ capacities regarding trauma and engagement with young in RC. In fact, this past can be a limitation when we work with the youth in these settings. Further studies are necessary to consolidate the results and identify any new factors that allow for or result from the participation of families in RC.

As expected, statistically significant differences were observed in setting objectives regarding the family’s participation in the LP. Thus, young people whose families participate in their LPs seem more actively involved in structuring future goals than those whose families do not participate in this process. According to the literature, young people’s experiences in RC tend to be marked by discontinuities that limit their future planning [8]. However, their families’ participation in their LPs may indicate a reorganization in creating or maintaining affective bonds, fostering greater internal availability of young people and, consequently, the delineation of life goals [15]. Therefore, parental figures involved in the RC process can prove to be preponderant sources of support and motivation, reinforcing the likelihood that young people become involved in the construction and implementation of LPs in a positive and targeted way [16]. These results were confirmed by Xu et al. [18], who state that integrating families in the transition of young people to RC through visits or intervention plans increases satisfaction and promotes the collaborative design of life goals between youth and their families.

Considering the conceptual model of the present study, the positive role of family support (autonomy) in setting goals was verified. Thus, it is suggested that the perception of the family as a welcoming nucleus that enhances autonomy mechanisms leads to a greater internal predisposition of young people to elaborate life goals. This result aligns with the study developed by Li and Cheng [37], which argues that the perception of affective security and parental autonomy in adolescents contributes to their involvement and persistence in defining and achieving goals. Thus, the encouragement of independence on the part of primary caregivers seems to be associated with processes of individuation, personal growth, independence, and self-determination [39,77,78], which, by extension, contribute to the greater involvement and security of young people in the planning of life goals [30,37]. However, given the dysfunctionality that often guides the parental care of young people living in RC, we cannot neglect the limiting effect that family can have on defining young people’s goals [79]. In this sense, we emphasize the importance of working with families from multiple points of view (e.g., (re)establishing positive relationships, developing a care/parenting strategy, ongoing support, and guidance), providing tools that foster in young people’s feelings of autonomy, individuation, and self-respect [80].

Indeed, the results indicated that the autonomy of family support is positively associated with resilience. Research has revealed that while promoting autonomy and security, family dynamics tend to help adolescents develop more positive perceptions of themselves and the world, increasing their internal resources to overcome challenging events [1]. Notably, in the context of RC, although the role of family in the resilient development of young people is still controversial, the systematic review developed by Pinheiro et al. [28] seems to be in line with the results presented, noting that family relationships, when worked on, can contribute to the positive adaptation of young people. This result suggests the need to implement more intervention programs and produce additional scientific knowledge about the role of parental figures in the adaptive development of the RC population.

The results also demonstrated associations between resilience and the goal-setting process. In this way, resilient young people can develop a more positive image of themselves, which could translate into greater motivation, security, and confidence in projecting their achievements in the future [8,53,56]. Using a comprehensive literature review, some authors found that resilience enables acquiring resources and tools to define more promising life trajectories (e.g., [8]).

Finally, the results allowed us to observe the indirect effects between family support (autonomy) and the goal-setting process since a positive mediating effect of resilience was verified in the previous association. According to the literature, social relationships, particularly relationships established with parents, play an important role in the way young people develop their well-being and autonomy [36]. Some researchers have highlighted that the family’s approach to supporting autonomy and independence contributes to consolidating young people’s personal identity, resulting in the definition of a purpose in life [38,39]. In this sense, it was found that the ability to respond to support the autonomy of primary caregivers can encourage resilient and adapted trajectories in young people living in RC, giving them more stability and resources to invest in developing life goals. This result reflects the importance of bonding experiences with primary care figures in the adaptive development of young people in RC. However, it is important to highlight that although the present study recognizes the relevance of family support in developing objectives, this reality only sometimes occurs. Notably, in the RC environment, family dynamics can be characterized by some instability [9], negatively impacting the developmental process of young people [81]. To this extent, we believe it is important to point out that despite having negative affective experiences with biological families, in light of attachment theory [1], young people are capable of developing new relationships with significant figures of affection who are capable of contributing to their process resilience and, therefore, development of life goals [5,82]. Thus, continued investment in working with RC professionals becomes relevant, fostering the development of affective relationships between young people and caregivers and building bridges with families.

The variables including age of entry into RC, length of stay in RC, and type of LP were also controlled in the model. The results made it possible to verify a negative association between reception time and the process of establishing objectives. Despite the scarcity of studies focusing on the duration of stays in RC and the inconsistency in the reported results [46,50,51,52], some studies suggest that prolonging the time spent in RC appears to limit young people’s aspirations and life opportunities [43,83]. These results may be because families do not always have a constant and consistent presence throughout young people’s experiences in RC. Inconsistencies in family dynamics combined with difficulties in building relationships with foster care professionals and a perception of a negative social climate in RC facilities could result in the loss of feelings of belonging and identity, limiting the outlining of objectives for young people in RC [52,83]. Considering the results obtained, it is important to discuss the excessive reception length of the present sample (*M* = 3.34; *SD* = 3.45 years), which is in line with the Portuguese reality. Furthermore, the significant number of young people whose current RC does not refer to the only one in which they were sheltered is worth mentioning. In this sense, we intend to highlight that the LPs of young people in RC may not be fully adjusted to their needs since reducing reception levels and promoting alternative measures (e.g., family care) still seem distant [14]. These data are corroborated by the report responsible for characterizing the RC situation in Portugal (CASA), which stated that the absolute number of LP reset needs appears to have increased considerably over the years [14], suggesting that adjustments in implementing these measures are imperative. Furthermore, the results draw attention to the possibility that interventions carried out with families do not have the desired effects, which promotes, because of their inability to care, successive returns of young people to the system and extensions of their reception measures. This reality is of particular concern when considering that entry into RC facilities tends to occur later, which could indicate that young people tend to remain in the care of disorganized, possibly negligent, and emotionally uncontainable family structures for more extended periods.

Although the construction of objectives has been associated with the age of entry into RC [43], the results of the present study did not reveal significant effects among the variables. According to the literature, the contexts in which individuals find themselves can shape their development over time [84]. Systematic exposure to contexts of neglect and abuse during childhood conditions young people’s opportunities [8], giving them lower expectations in different dimensions of life [35]. However, contact with new, welcoming, and receptive relational contexts increases young people’s involvement in creating and achieving goals [82]. As such, it appears that young people tend to set their life goals based on the limitations (e.g., intensity and commutative effect of adverse experiences) and opportunities (e.g., construction of safe alternative relationships) provided by the environment [15,31,32], meaning that the age at entry into RC by itself does not show any direct contribution to these processes.

Finally, the results did not suggest an effect of LP typology on young people’s goal-setting processes. Despite the high scarcity of studies on the subject, Brites et al. [85] mention that young people perceive opportunities to participate in LP design as a factor impacting their motivation to plan and achieve life goals. However, young people’s involvement in defining LPs is only sometimes observed in RC contexts [7,86], and many young people need to be completely aware of the implications of LPs. Therefore, young people’s ability to set goals may be independent of the LP in which they are inserted. In the present sample, other variables related to institutional and family functioning dynamics are more associated with establishing young people’s goals, namely, family support, family involvement in LPs, and length of stay in RC.

### Practical Implications, Limitations, and Future Directions

The present study contributes to advancing scientific knowledge, particularly about family support and its relationship with the life goals and resilience of the population in RC, by providing important practical implications for child and youth promotion and the protection system. This study’s results highlight the importance of family support in outlining young people’s goals. Furthermore, they highlight the family’s contribution to the positive adaptation of this population. In this sense, we hope that these conclusions will contribute to the development of policies and practices aimed at supporting and improving the relationship between parents and children in RC, as well as clarifying the importance of greater involvement of primary caregivers in this context (e.g., through participation in LP design) [87,88]. Additionally, because more and more people are entering RC at an advanced age [14], we propose investing in developing multidisciplinary intervention programs with families that allow them to increase their capacity to respond and develop parenting/care skills. Thus, the articulation between RC facilities and families is fundamental.

In addition, once it has been established that the extended stay of young people in RC makes it difficult for them to construct their life goals, continuous reinforcements and investments in the support, supervision, and training of RC professionals are recommended [50,88]. This recommendation arises from the significant role RC careworkers can play in developing young people’s life goals [82] and assumes particular relevance when depriving contact with families in RC is in the best interest of the children and young people. In this sense, it is expected that this research will promote greater quality assurance in the standards of services provided by RC, namely, through the prevention of turnover and emotional exhaustion of caregivers, the adequacy of young person/caregiver ratios, and the allocation of working hours and adequate compensation salaries for professionals [29,50]. It would also be interesting to develop interventions aimed at young people to strengthen their involvement in devising significant objectives during their stay in RC and after transitioning to independent life. Finally, this research intends to serve as a reference for advances in research around LPs in RC, highlighting the relevance of reinforcing the involvement of various figures such as teachers, RC staff, parental figures, and, particularly, young people in its definition and implementation.

Although the present study provides notable contributions to the advancement of research, it is important to recognize some limitations. Firstly, the exclusive use of self-report instruments must be highlighted, which are susceptible to bias because of the risk of socially desirable responses. It is important to highlight that this measure must involve special attention when evaluating family support, given the reluctance and sensitivity that this issue can have among the population in RC. Other areas for improvement of the present study include its cross-sectional design, which prevents the identification of cause–effect relationships and temporal monitoring of variables, as well as the small sample size and disproportionality of participants between both sexes. In addition to the above, the present study should have considered the unanimous distribution of young people in univariate analyses. Potential limitations may also be associated with the non-adaptation of the instruments for the target population and the fact that the objective setting variable is exclusively measured by one dimension of the SSQR. Finally, because of the fatigue factor, the protocol length may have affected the responses’ coherence and increased the number of non-responses.

Finally, we suggest that future lines of investigation should examine other variables that may influence the process of setting young people’s goals in RC (e.g., self-efficacy, quality of relationships with siblings, peers, teachers, and RC professionals). On the other hand, it would be interesting to identify other contextual variables related to young people’s family environments (e.g., the nature of the pre-care relationship, the reason for placement, format, and regularity of contacts). Including this information could expand the compression of family relationships during RC and their roles in young people’s life goals. Given the innovative nature of this study, it is crucial to promote more scientific research to clarify the impact of families’ participation in RC and the length of time spent in care on the formation of young people’s life goals in RC. Likewise, it would be valuable to develop longitudinal studies to establish causality in the relationship among the variables under study and monitor their development over time. Furthermore, future studies should monitor the design and implementation of LPs, including interventions aimed at family placement or (co)construction of young people’s autonomy. To conclude, including a qualitative methodology in future investigations is an important complement to this study to collect in-depth information about the affective experiences of young people, their adaptive processes, and resources for elaborating and achieving objectives. It would also be relevant to carry out semi-structured interviews aimed at significant figures, namely, parents, teachers, and RC professionals, or even consider the crossing of information provided by multiple informants. Adopting this approach will provide added value for building robust emotional and behavioral profiles of young people. Furthermore, it may give a broader view of the possible influence of these figures on the definition and implementation of projects and life goals in RC. A relevant point for intervention would be to introduce national cases to analyze different realities and promote specific policies related to the implementation of RC programs, including careworkers and youth families.

## 5. Conclusions

The results allow us to gain a broader understanding of emotional experiences, future projections and youth decision-making processes in RC. Additionally, this study allows the expansion of knowledge of the LP stipulated by the RC facilities in Portugal, highlighting the role of the family in defining and realising them. The results show that the presence of family relationships characterised by support, availability, understanding and affection, positively predicts the goal-setting of young people in RC. Family support based on the autonomy and independence of young people has no effect on the life goals they set for themselves. However, there was a positive and significant association between family support (adaptation) and goal-setting through the resilient process (acceptance of self and life). At the same time, it was found that short lengths of stays in RC organise a protective factor with regard to planning future goals. This study enabled us to learn about the importance of developing secure relationships with family in the resilience process of young people in RC. Although controversial, it provides knowledge about the importance of family support in the development of coping strategies and goals for the future of young people living in disadvantaged situations. More investment is needed to work on family intervention programmes in the context of RC.

## Figures and Tables

**Figure 1 behavsci-14-00581-f001:**
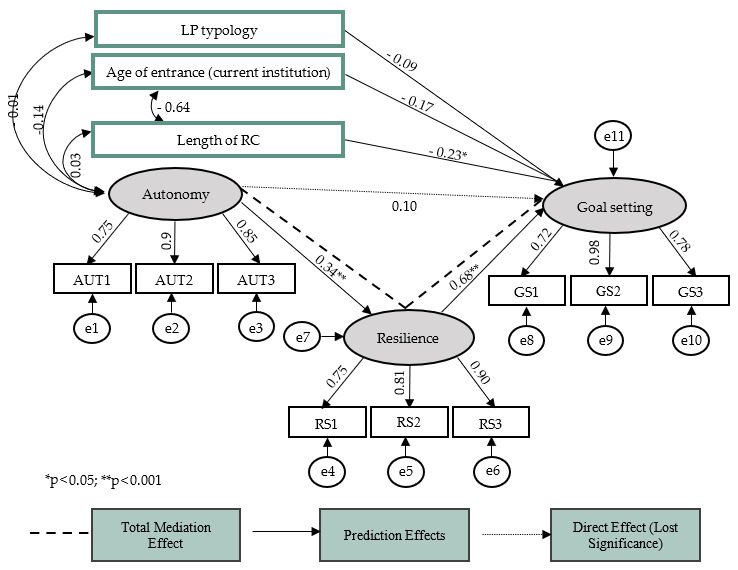
Representative model of the mediating effect of resilience on the association between family support and the goal-setting process.

**Table 1 behavsci-14-00581-t001:** Correlation among variables and mean and standard deviation (n = 124).

Variable	1	2	3	4
Family support				
1. Autonomy	-			
2. Adaptation	0.471 **	-		
3. Resilience	0.269 **	0.253 **	-	
4. Goal setting	0.307 **	0.274 **	0.631 **	-
*M*	2.97	3.18	5.35	4.00
*SD*	0.830	0.789	0.987	0.780

*M*, mean; *SD*, standard deviation; ** *p* < 0.01.

**Table 2 behavsci-14-00581-t002:** Differential analysis of family support, resilience, and goal setting according to sex.

Variable	Sex		IC95%	Direction of Significant Differences
	1. Male (n = 75) *M* ± *SD*	2. Female (n = 49) *M* ± *SD*		
Family support				
Autonomy	3.03 ± 0.832	2.88 ± 0.826	[−0.156; 0.447]	n.s.
Adaptation	3.32 ± 0.796	2.97 ± 0.739	[0.006; 0.629]	1 > 2
Resilience	5.54 ± 0.897	5.06 ± 1.056	[0.130; 0.830]	1 > 2
Goal setting	4.15 ± 0.785	3.77 ± 0.718	[0.111; 0.664]	1 > 2

*M*, mean; *SD*, standard deviation; IC, confidence interval; n.s., non-significant.

**Table 3 behavsci-14-00581-t003:** Differential analysis of family support, resilience, and goal setting according to family participation in the LP.

Variables	Family Participation in the LP		IC95%	Direction of Significant Differences
	1. Yes (n = 60) *M* ± *SD*	2. No (n = 35) *M* ± *SD*		
Family support				
Autonomy	3.15 ± 0.802	2.91 ± 0.864	[−0.108; 0.589]	n.s.
Adaptation	3.46 ± 0.631	2.73 ± 0.891	[0.380; 1.067]	1 > 2
Resilience	5.65 ±0.933	5.17 ± 0.921	[0.091; 0.875]	1 > 2
Goal setting	4.17 ± 0.764	3.81 ± 0.825	[0.035; 0.700]	1 > 2

*M*, mean; *SD*, standard deviation; IC, confidence interval; n.s., non-significant.

## Data Availability

Data are contained within the article.

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
