# Peer review of "Family Support, Resilience, and Life Goals of Young People in Residential Care"

_behavsci, 2024, doi:10.3390/bs14070581_

Round 1
Reviewer 1 Report
Comments and Suggestions for Authors
The paper is very well written and engaging. The methodology is carefully detailed and the conclusions are interesting with new lines of research and intervention, assuming the limitations found in the present study.
As a suggestion for improvement, and based on my own experience as a researcher, I would recommend using JAMOVI software (based on R-commander) for the adjustment of the structural equation model, as it tends to vary in the adjustment proposed by AMOS and in many cases presents a better accuracy.
Author Response
REVIEWER 1
- The paper is very well written and engaging. The methodology is carefully detailed and the conclusions are interesting with new lines of research and intervention, assuming the limitations found in the present study.
Authors: Thank you for taking the time to read our work.
- As a suggestion for improvement, and based on my own experience as a researcher, I would recommend using JAMOVI software (based on R-commander) for the adjustment of the structural equation model, as it tends to vary in the adjustment proposed by AMOS and in many cases presents a better accuracy.
Authors: Thanks. We accept the suggestion for future studies. The present work has been developed according to the mastery of the main author's programme, which we believe has given us precise and rigorous results.

Reviewer 2 Report
Comments and Suggestions for Authors
Generally, the article is well structured and of great interest, although some minor changes are suggested.
When differences among male and female youth is discussed, it would be preferable to use always or "gender" or "sex".
In "participants" section, a brief description of RC facilities could be provided to the reader, especially if they differ in some way (age range of guests, type of additional services...).
Results section is well detailed, all data analysis techniques are clearly explained.
Discussion section is coherent with RQs and resutls. Although I found this sentence "This evidence seems to be in line not only with traditionally assigned gender roles but also with differences in the search for closeness by individuals of both sexes" not clear enough. I suggest to rephrase it.
Considering the discussion on the limits of involving families in LP of youth, due to instability and negative influence they may have sometimes, this aspect should be anticipated more clearly in the introduction section, by discussing more in depth relevant literature, and by justifying better the topic of study. I also suggest to consider to add few words in methods section about the reasons why other significant adult figures haven't been considered in the study, when families are not able to support positively young people. This aspect is well addressed in section 4, especially 4.1 and study limits, but it should be discussed also previously (although I aknowledge that it is not among the objectives of the study).
Political implication should be better addressed, as the research presented seems to have lots more of possible practical application; I suggest to contextualize dealing more with the national case presented, as policies, RC programs, culture, and family structures are mostly context dependent. This could give more relevance to the study presented.
I suggest to check reference 90, as I could not find it in the text.
Comments on the Quality of English LanguageThe article is well written, no issues about language and sintax.
Author Response
REVIEWER 2
Generally, the article is well structured and of great interest, although some minor changes are suggested.
- When differences among male and female youth is discussed, it would be preferable to use always or "gender" or "sex".
Authors: Thanks for the comment; we accept the suggestion, and we review the discussion considering the sex (that was what we asked for). However, we maintain that some points of gender expression to explain the results of the present study seem to be justified by cultural and gender stereotypes.
- In "participants" section, a brief description of RC facilities could be provided to the reader, especially if they differ in some way (age range of guests, type of additional services...).
Authors: Thanks for the suggestion; as requested, we introduce a brief characterisation of the institutions to clarify what kind of situations led young people to residential care. We have also added information on situations excluded from the sample.
Page 3. It should be noted that the institutions participating in the present study did not include specialized/therapeutic RC. These adolescents lived in a residential care institution due to a diverse set of adverse life situations, namely parental neglect or lack of family socio-economic conditions. The participating residential care institutions did not include children with mental disabilities/disorders or deviant behaviors (conduct disorders or substance abuse). The sample included in this study is homogeneous in relation to race and ethnicity.
- Results section is well detailed, all data analysis techniques are clearly explained.
Authors: Thanks for the Comment.
- Discussion section is coherent with RQs and resutls. Although I found this sentence "This evidence seems to be in line not only with traditionally assigned gender roles but also with differences in the search for closeness by individuals of both sexes" not clear enough. I suggest to rephrase it.
Authors: According to your suggestion we clarify the sentence.
Page 9. This evidence seems to be in line with traditionally assigned gender roles and also with the differences in proximity-seeking by individuals of both sexes, whereby proximity-seeking is more prominent in females.
- Considering the discussion on the limits of involving families in LP of youth, due to instability and negative influence they may have sometimes, this aspect should be anticipated more clearly in the introduction section, by discussing more in depth relevant literature, and by justifying better the topic of study.
Authors: We agree with the suggestion. Although this discussion has already been launched in the introduction, it is important, despite the controversy, to reinforce the idea of family participation in working with young people
Page 3. Therefore, considering that young people in RC experience inconsistent care and affective discontinuities in the family [6, 42] but at the same time continue to feel a sense of belonging to the primary caregivers, it is important to work on family dynamics with young people and the institution, in order to promote parental involvement and re-education for young people's adaptive life projects [7,8].
- I also suggest to consider to add few words in methods section about the reasons why other significant adult figures haven't been considered in the study, when families are not able to support positively young people. This aspect is well addressed in section 4, especially 4.1 and study limits, but it should be discussed also previously (although I aknowledge that it is not among the objectives of the study).
Authors: Thanks for the suggestion; we added some notes.
Page 3/4. Although other significant adult figures may be relevant to young people who experience affective discontinuities (e.g. friends, godparents, coaches), in this study, we emphasise the role of the family, which is usually the one that maintains contact with young people, and need more attention on young people emotional involvement.
- Political implication should be better addressed, as the research presented seems to have lots more of possible practical application; I suggest to contextualize dealing more with the national case presented, as policies, RC programs, culture, and family structures are mostly context dependent. This could give more relevance to the study presented.
Authors: Thank you, the comments make sense. Although we had already mentioned some practical implications and future directions related to intervention programs, we added some of the suggested ideas.
Page 14. A relevant point for intervention would be to introduce national cases to analyse different realities and promote specific policies related to the implementation of RC programs, including careworkers and youth families.
- I suggest to check reference 90, as I could not find it in the text.
Authors: Thanks. We reorganize references due to some error, and the nº 90 is now nº 89 in the text.
- Comments on the Quality of English Language: The article is well written, no issues about language and sintax.
Authors: Thanks for the comment.

Reviewer 3 Report
Comments and Suggestions for Authors
The article has a quantitative design. As a qualitative researcher, I cannot therefore assess the validity of the analyses. The article is interesting and provides valuable insights about family support to people in residential care. The study's results highlight the importance of family support to young people’s goal. The article can contribute to awareness of the importance of strengthening the relationship between parents and children in residential care. The way I read the article, it appears to be important and well-prepared, and the analyses are credible.
Author Response
The article has a quantitative design. As a qualitative researcher, I cannot therefore assess the validity of the analyses. The article is interesting and provides valuable insights about family support to people in residential care. The study's results highlight the importance of family support to young people’s goal. The article can contribute to awareness of the importance of strengthening the relationship between parents and children in residential care. The way I read the article, it appears to be important and well-prepared, and the analyses are credible.
- Authors: Thank you for taking the time to read our work.

Reviewer 4 Report
Comments and Suggestions for Authors
Page 1: You do a good job of introducing the basics of attachment theory and defining your core terms/abbreviations.
P2: Appreciate attention to gender details and discussion of different caregiver potentials. Good discussion of connection between positive, supportive authority in an environment and ability to create life goals.
p3: Good discussion of ecological factors.
2.1: Good description of participants with overview of LP data.
2.2: Thorough relation of questions and measurement standards
2.3: repeat "informed" consent twice.
P6: Could indicate that greater sexist standards may determine differences in adaptability and goal setting (although the explanation on page 9 is quite thorough). Rest of page 6 makes sense.
P8: Summary at bottom of page quite helpful.
P9: Great discussion of perception of support by gender and explanation for discrepancy.
P10: May want to add an additional note that the repair potential offered for parents acknowledging issues in child's earlier relationship can provide great healing potential. Also, addressing why studies are limited (emphasizing, again, the correlation with past trauma in childhood often associated with nuclear families) and how this factor is accounted for during visits in RC, may help to allay any reader's concerns. You allude to this at the bottom of the page in relation to potential harmful factors, but it may be helpful to be explicit at the top of the page that this potential is a reason for limited studies so far.
Page 12: Caveat of the exceptional importance of caregirvers when families aren't healthy is well-done and vital.
P13: Good discussion of limits of self-reporting and directions for future research.
Author Response
REVIEWER 4
Page 1: You do a good job of introducing the basics of attachment theory and defining your core terms/abbreviations.
Authors: Thanks for the comment.
P2: Appreciate attention to gender details and discussion of different caregiver potentials. Good discussion of connection between positive, supportive authority in an environment and ability to create life goals.
Authors: Thanks for the comment.
p3: Good discussion of ecological factors.
Authors: Thanks for the comment.
2.1: Good description of participants with overview of LP data.
Authors: Thanks for the comment.
2.2: Thorough relation of questions and measurement standards
Authors: Thanks for the comment.
2.3: repeat "informed" consent twice.
Authors: Thank you for your suggestion; we removed “informed” once.
P6: Could indicate that greater sexist standards may determine differences in adaptability and goal setting (although the explanation on page 9 is quite thorough). Rest of page 6 makes sense.
Authors: thanks for the comment; we think that we have already pointed out the sexist standards and cultural implications about the gender roles associated with sex. You can point it out in the introduction. However, we introduce additional information to reinforce this position.
Page 2. Despite the controversy of the results reported by the scientific community, gender effects are found on the behavioural adjustment of adolescents living in RC [46]. Boys tend to adopt more reactive, violent, and deviant behaviours compared to girls; on the other hand, girls tend to report more internalizing problems such as depression and anxiety, which in both cases can affect the youth´s adaptation and the performance setting [23,63, 66, 69].
P8: Summary at bottom of page quite helpful.
Authors: As suggested, we add a summary.
P.8 Concluding, the results stress the importance that having “strengthened family will be able to fulfil its functions and tasks, which will in turn contribute to the strengthening of individual family members as well as the community in which family is living” [4] (p. 15).
P9: Great discussion of perception of support by gender and explanation for discrepancy.
Authors: Thanks for the comment.
P10: May want to add an additional note that the repair potential offered for parents acknowledging issues in child's earlier relationship can provide great healing potential. Also, addressing why studies are limited (emphasizing, again, the correlation with past trauma in childhood often associated with nuclear families) and how this factor is accounted for during visits in RC, may help to allay any reader's concerns. You allude to this at the bottom of the page in relation to potential harmful factors, but it may be helpful to be explicit at the top of the page that this potential is a reason for limited studies so far.
Authors: Dear authors thank you for your suggestions. We added some of the suggested ideas.
Page 10: Pinheiro et al. [29] position that past trauma experiences cannot be “changed” it is important to develop parents’ capacities regarding trauma and engagement with young in RC. In fact, this past can be a limitation when we work with the young in these settings.
Page 12: Caveat of the exceptional importance of caregivers when families aren't healthy is well-done and vital.
Authors: Thanks for the comment.
P13: Good discussion of limits of self-reporting and directions for future research.
Authors: Thanks for the comment.
